# Resistance Training Acutely Impairs Agility and Spike-Specific Performance Measures in Collegiate Female Volleyball Players Returning from the Off-Season

**DOI:** 10.3390/ijerph17186448

**Published:** 2020-09-04

**Authors:** Kenji Doma, Jonathan Connor, Daniel Gahreman, Daniel Boullosa, Juha P. Ahtiainen, Akinori Nagata

**Affiliations:** 1College of Healthcare Sciences, James Cook University, Townsville 4811, Australia; jonathan.connor@jcu.edu.au (J.C.); daniel.boullosa@gmail.com (D.B.); 2College of Health and Human Sciences, Charles Darwin University, Darwin 0909, Australia; Daniel.Gahreman@cdu.edu.au; 3INISA, Federal University of Mato Grosso do Sul, Campo Grande 79070-900, Brazil; 4Neuromuscular Research Center, Faculty of Sport and Health Sciences, University of Jyväskylä, 40014 Jyväskylä, Finland; juha.ahtiainen@jyu.fi; 5Faculty of Social Welfare, Rissho University, Kumagaya 360-0194, Japan; akinori@ris.ac.jp

**Keywords:** strength training, muscle damage, muscle soreness, jump, change-of-direction

## Abstract

This study examined the acute effects of resistance training (RT) on volleyball-specific performance. Sixteen female volleyball players undertook their initial, pre-season RT bout. Countermovement jump (CMJ), delayed onset of muscle soreness (DOMS), and sport-specific performances (i.e., run-up jump, agility, and spiking speed and accuracy) were measured before, 24 (T24), and 48 (T48) hours after RT. A significant increase in DOMS was observed at T24 and T48 (~207.6% ± 119.3%; *p* < 0.05; ES = 1.8 (95% CI: 0.94–2.57)), whilst agility was significantly impaired at T48 (1.7% ± 2.5%; *p* < 0.05; ES = 0.30 (95% CI: −0.99–0.40)). However, there were no differences in CMJ (~−2.21% ± 7.6%; *p* > 0.05; ES = −0.11 (95% CI: −0.80–0.58)) and run-up jump (~−1.4% ± 4.7%; *p* > 0.05; ES = −0.07 (95% CI: −0.76–0.63)). Spiking speed was significantly reduced (−3.5% ± 4.4%; *p* < 0.05; ES = −0.28 (95% CI: −0.43–0.97)), although accuracy was improved (38.3% ± 81.4%: *p* < 0.05) at T48. Thus, the initial, preseason RT bout compromised agility and spiking speed for several days post-exercise. Conversely, spiking accuracy improved, suggesting a speed–accuracy trade-off. Nonetheless, at least a 48-h recovery may be necessary after the initial RT bout for athletes returning from the off-season or injury.

## 1. Introduction

Volleyball is a team sport involving high-intensity, intermittent bouts of different activities, interspersed with longer recovery periods [1]. These repeated, high-intensity bouts involve maximal accelerations, jumping, and change-of-direction movements in order to successfully receive, block, and spike the ball during a match [2]. Thus, volleyball players are commonly required to undertake resistance training (RT) to develop force–velocity characteristics of performance [3]. However, RT can also cause exercise-induced muscle damage (EIMD), largely in those unaccustomed to this mode of training [4,5]. The common symptoms of EIMD include muscle soreness, increased leakage of intramuscular proteins into the systemic circulatory system (e.g., creatine kinase (CK)), and muscle stiffness and attenuation of muscular contractility for 24–72 h post-exercise [6,7,8]. Whilst not fully understood, it has been proposed that strenuous, eccentric activity may cause losses in force production, due to damage to the sarcomeres and impaired excitation–contraction coupling process [9]. In this regard, athletes returning from injuries, or the off-season, who are temporarily deconditioned may be susceptible to symptoms of EIMD following the initial RT bout [10]. For example, a recent study reported that an RT bout caused high levels of EIMD symptoms in individuals previously exposed to RT, following 4 weeks of deconditioning [11]. In addition, resistance-trained individuals have been shown to experience EIMD symptoms following muscle-damaging protocols [12,13], demonstrating that EIMD may not be completely avoidable, irrespective of training background, with greater susceptibility of EIMD following longer periods of deconditioning [11]. Thus, recovery dynamics should be acutely monitored following RT, as presence of EIMD may compromise training quality, or increase susceptibility of injuries [14].

Studies have reported high levels of EIMD with concomitant impairments in running economy [15,16,17], running time-trial performance [18], and cycling performance [19] for several days following a bout of resistance exercises. These findings suggest that a bout of RT may impair the quality of subsequent endurance training sessions as a result of EIMD symptoms [10]. However, running economy and cycling performance are not indicative of physical demands in volleyball, which requires repeated, high-intensity activities involving sprinting, change-of-direction speed, and spiking (i.e., hitting the ball) [2].

More recently, a study by de Freitas et al. [20] examined the acute effects of five days of intensified training in a sample of six elite, male volleyball players. The results of this preliminary study [20] demonstrated that the training protocol augmented indirect muscle damage markers (i.e., muscle soreness, increased serum CK activity, and thigh circumference), although surprisingly, squat jump and agility performances, which are considered important measures for volleyball [1,2], were not affected. These findings suggest that determinants of volleyball performance measures may be unaffected during periods of EIMD in elite, male volleyball players. However, the training protocol was incorporated once athletes resumed training during the pre-season, suggesting that athletes may have already been conditioned for high levels of EIMD, based on the repeated bout effect [21]. Examining the initial bout of RT returning from the off-season may induce greater levels of EIMD, thereby increasing the possibility of impairing physical measures pertinent for volleyball. Furthermore, symptoms of EIMD have been demonstrated to be distinct between sexes [22], and thus, acute responses of an RT session may differ in female volleyball players. Finally, no studies have examined spiking ability (i.e., ball speed and accuracy) during periods of EIMD as far as we are aware, despite this task being considered paramount during volleyball matches [1,2,23]. Therefore, the current study examined the recovery from a bout of typical, whole body RT on determinants of volleyball performance in elite, female volleyball players returning from the off-season. It was hypothesized that a bout of RT will cause EIMD, and concomitantly impair volleyball-specific performance measures in elite, female volleyball players deconditioned due to the off-season.

## 2. Materials and Methods

### 2.1. Research Design

This study was conducted as a repeated measures design across two weeks. The first week consisted of a familiarization session and a baseline assessment session. During the familiarization session, the athletes were acquainted with all protocols incorporated in the study, and a one-repetition maximum assessment was conducted for all the resistance exercises performed during the subsequent week. The baseline assessment session (TBase) consisted of indirect muscle damage markers, vertical jump, agility performance, and spike-specific performance protocols. During the second week, at least three days following TBase, a whole-body RT bout was undertaken, followed by assessment sessions equivalent to that of the baseline assessment session at 24 h (T24) and 48 h (T48) after the RT bout. To minimize potential influence of exercise on EIMD, no training was undertaken for up to 48 h following the RT bout.

### 2.2. Participants

The participants were 16 female college volleyball players (age 21 ± 1 years; height 1.71 ± 0.05 m; body mass 77.8 ± 12.6 kg) who competed in national level competitions. All participants played volleyball in highly competitive events for 10.2 ± 2.7 years and had 5.5 ± 2.1 years of experience undertaking RT. At the time of the study, they were returning from a 3-week off-season. During the off-season, the participants continued undertaking volleyball-specific conditioning (e.g., spiking, receiving, sprinting) 1–2 times per week for three weeks, but either refrained from a structured RT program, or substantially reduced their RT volume by at least half, as typically instructed by volleyball coaches for off-season recovery. Once participants received the general background of the study, testing procedures, and potential risks involved, they provided informed consent. All experimental procedures were approved by the Experiment Ethics Committee of Chukyo University (approval number: 2019-13), and were conducted in accordance with the Declaration of Helsinki.

### 2.3. Indirect Muscle Damage Markers

The indirect muscle damage markers included delayed onset of muscle soreness (DOMS) of specific muscle groups (i.e., quadriceps (DOMSQuad), gluteal muscles (DOMSGts), triceps (DOMSTri), abdominals (DOMSAbs), and squat soreness (DOMSSqt)). The DOMS for each muscle group was assessed with the following technique: DOMSQuad—isometric contraction of the right quadricep muscles with the knee in an extended position; DOMSGts—isometrically contracting the gluteal muscles in a standing position; DOMSTri—performance of one push-up; DOMSAbs—performance of one sit-up; and DOMSSqt—overall lower body soreness during one bodyweight squat. The level of DOMS was assessed using a visual analogue scale from 1–10, with 1 defined as “no soreness” and 10 as “very, very sore” [15,21]. The lower body range of movement was assessed using a standard sit-and-reach (SR) test [24].

### 2.4. Sport-Specific Jump and Agility

The sport-specific performance measures included the countermovement jump (CMJ) test, run-up jump protocols (VJRUN) and a T-agility test protocol. The CMJ test was conducted with two maximal attempts, interspersed using a custom-built vertical jump apparatus (Chukyo University, Japan). This vertical jump apparatus consisted of a wooden pole screwed to a wall, attached with pegs separated by 1 cm. Participants were provided with 30–60 s of rest in-between each CMJ attempt, and the best score was recorded for later analyses. To ensure standardization, the participants were instructed to maintain their heels in contact with the ground during the eccentric phases prior to take-off, and to come off the floor with their knees and ankles fully extended, sustaining a correct posture and body alignment throughout the movement. The VJRUN was completed by having participants run towards the vertical jump apparatus from a 3 m distance, leaving the floor with two feet (i.e., bilateral jump) and tapping the pegs with the hand used for spiking the ball [25]. Two attempts were provided for the VJRUN, interspersed with approximately 30–60 s of rest, and the best score was recorded for later analyses. The VJRUN was included to measure jump height during a task that replicated a maneuver when spiking the ball during a volleyball match, with excellent reliability as previously reported [25]. The T-agility test was conducted in a 10 m × 10 m perimeter, with excellent reliability, as previously reported [26]. To complete the agility course, the participants sprinted to the center cone (cone A), sidestepped to the right cone (cone B), sidestepped to the far-left cone (cone C), sidestepped back to the center cone (cone A), and then back-pedaled to the starting line. To ensure familiarity with the protocol, the participants jogged around the cone at approximately 50% of maximum effort, followed by approximately 90% of maximum effort prior to recording the times. Two minutes after this warm-up, the participants completed two attempts at maximum effort, separated by at least 2 min, with the best time recorded for later analyses. The completion time for each trial was recorded using an electronic timing gate system (Swift Performance, Brisbane, Australia), which was stationed at the starting line.

### 2.5. Spike-Specific Performance

The spike-specific performance test was undertaken on a standard volleyball court (9 m × 18 m) and the net height was set for female volleyball players (2.24 m) [27]. As part of the assessment, participants were instructed to spike the ball at a target of 6 m × 3 m dimension (Figure 1). The participants were allowed five attempts, and each attempt was scored with a dichotomous parameter of “successful” or “unsuccessful”. The participants were not provided with extra attempts, irrespective of the quality of each set. The ball speed for each spike was also measured with a speed gun (Stalker ATS II, Texas, USA), at a sampling rate of 100 Hz. The same setter passed the ball to the participant to ensure standardization across each trial. According to a previous study [27], this spike-specific performance protocol has been reported with excellent reliability in a similar cohort of female volleyball players.

### 2.6. Repetition Maximum Assessment

The participants commenced with a progressive warm-up by jogging at a self-selected pace for 2–3 min, followed by dynamic stretches of the upper and lower extremities (i.e., leg and arm swings, body weight squats, and alternating lunges). After the warm-up, the 1RM tests were conducted in the following order: power cleans, deadlifts, seated dumbbell shoulder press, and back squats, as in previously-described methods [21]. Specifically, the participants undertook warm-up sets of 10 repetitions of each exercise at approximately 50–60% of their previously recorded 1RM, followed by 2–3 repetitions at 75–80% of 1RM. After 2–3 min of recovery, participants attempted their 1RM for each exercise. Loads were altered by 5–15% if participants perceived the load to be excessively light or heavy, with approximately 5 min of rest provided in-between each attempt and exercise.

### 2.7. Resistance Training Bout

The RT bout consisted of upper and lower body exercises, which were performed in order: power cleans (three sets of five repetitions at 80% of 1RM), deadlift (five sets of five repetitions at 80% of 1RM), dumbbell shoulder press (three sets of ten repetitions at 70% of 1RM), back squats (five sets of five repetitions at 80% of 1RM), and dumbbell lunges (four sets of ten repetitions on each leg at ~40% of body mass). Approximately two minutes of rest was provided in-between each set of each exercise. If participants perceived that they were able to undertake more of the assigned repetitions in a set, or were unable to complete the assigned repetitions in a set, the load was adjusted by 2–5% for the subsequent working sets.

### 2.8. Statistical Analysis

The continuous measures were reported as mean ± standard deviation, and the accuracy of the spiking protocol (i.e., successful versus unsuccessful) was reported as frequencies via a cross-tabulated table. Data normality was checked using the Shapiro–Wilk test, and then analyzed using the Statistical Package of Social Sciences (SPSS, IBM, v25). A two-way (day × spike) repeated measures analysis of variance (ANOVA) was used to compare ball speed of the volleyball-specific performance protocol between testing days (i.e., TBASE, T24, and T48) and the five spiking attempts on each day. A one-way repeated measures ANOVA was then conducted for DOMS and SR and the sport-specific performance measures (i.e., CMJ, VJRUN, and agility). For post-hoc analyses of the repeated measures ANOVA, Bonferonni’s pairwise comparison was employed. Effect size (ES; Cohen’s d) with 95% confidence interval (CI) was also calculated to examine the magnitude of differences in the dependent measures for interaction and main effects, with 0.2 considered as a small ES, ≥0.5 as a moderate ES, and ≥0.8 as a large ES [28]. To compare the success rate of the spiking protocol, a McNemar test was used between the days (i.e., TBASE, T24, and T48). The alpha level was set at 0.05 for all analyses.

## 3. Results

### 3.1. Indirect Muscle Damage Markers

Significant increases in all DOMS measures were also observed at T24 and T48 when compared to TBase (*p* < 0.01; Table 1) with large ES (1.47–2.57; Table 2), although no differences were found between T24 and T48 (*p* > 0.05) with small ES (0.02–0.47; Table 2).

### 3.2. Sport-Specific and Spike-Specific Performance Measures

The CMJ exhibited no differences at T24 (*p* > 0.05; Figure 2) and T48 (*p* > 0.05) when compared to TBase with small ES (0.08 and 0.14, respectively), and between T24 and T48 (*p* > 0.05) with a small ES (0.07). For VJRUN, no significant differences were found at T24 (*p* > 0.05; ES = 0.05 (95% CI: −0.75–0.64); Figure 3) and T48 (*p* > 0.05; ES = 0.08 (95% CI: −0.77–0.62)) when compared to TBase, and between T24 and T48 (*p* > 0.05; ES = 0.02 (95% CI: −0.71–0.67)). However, time-to-completion for the T-agility test significantly increased at T48 when compared to TBase (*p* = 0.04; ES = 0.30 (95% CI: −0.99–0.40); Figure 2), although no differences were found between TBase and T24 (*p* > 0.05; ES = 0.10 (95% CI: −0.80–0.59)) and between T24 and T48 (*p* > 0.05; ES = 0.20 (95% CI: −0.89–0.50)). There was no interaction effect for the ball speed during the spike-specific performance protocol (*p* = 0.57; Figure 3), with no main effects between spiking attempts (*p* = 0.11). However, a main effect was found between days (*p* = 0.04; Figure 3), with significantly slower ball speed at T48 (58.0 ± 7.7 km·hr^−1^) when compared to TBase (60.0 ± 6.7 km·hr^−1^; *p* = 0.02; ES = (95% CI: −0.43–0.97)). For the accuracy of the spike-specific performance protocol, there was a significantly greater number of successful attempts at T48 (87.5%) when compared to TBase (73.8%; *p* = 0.03), although the participants exhibited a similar number of successful attempts between TBase and T24 (78.8%; *p* = 0.60) and between T24 and T48 (*p* = 0.19).

## 4. Discussion

The current study shows that the present RT bout-induced DOMS of the upper, trunk, and lower body and impaired several movement demands typically observed during a volleyball match (i.e., agility and ball speed during a spike-specific performance task). However, no differences were found in vertical jump performance measures, whilst improvements were observed in spike accuracy. Accordingly, sufficient recovery should be considered when incorporating court-specific conditioning exercises that involve change-of-direction movements and spiking drills following the initial RT bout for elite, female volleyball players returning from the off-season.

In the current study, the presence of DOMS is particularly interesting, given that the participants were resistance-trained with only three weeks of deconditioning as a result of the off-season. As far as we are aware, no studies have acutely monitored indirect muscle damage markers in elite, resistance-trained athletes returning from the off-season. Whilst it has been reported that individuals exposed to RT are protected from symptoms of EIMD for 6–9 months post-exercise due to the repeated bout effect [29], our findings show that three weeks of absence from, or reduced volume of, RT is sufficient to cause low-to-moderate levels of EIMD in highly resistance-trained athletes. Previously, Hassan [11] examined EIMD symptoms across two bouts of RT, with the two RT bouts separated by two weeks in one group, and by four weeks in another group. The results showed that the group who undertook the two RT bouts across two weeks exhibited attenuated symptoms of DOMS following the second bout of resistance exercises. However, this repeated bout effect phenomenon was not observed for the other group, with comparable levels of elevated DOMS between the resistance exercise bouts. Thus, our findings are in accordance with the results by Hassan [11], whereby several weeks of deconditioning may increase susceptibility of RT-induced DOMS.

Whilst DOMS was elevated as a result of the RT bout, CMJ and VJRUN were unperturbed in the current study. These findings were unexpected, given that symptoms of DOMS have been shown to impair muscular contractility [30,31]. Furthermore, the current findings differ to those by previous studies, with impaired jumping capabilities during periods of EIMD as a result of lower body RT for 48 h post-exercise [21,32]. The discrepancy in findings between the current study, and those by others [21,32] may be due to distinct familiarity of jumping task constraints, or differences on methods of evaluation. For example, studies that have reported compromise in jump height for several days following an RT bout were moderately-trained runners [21,32], with no RT backgrounds. In comparison, the participants in the current study were highly anaerobically-trained, with exceptional jumping capabilities due to the requirement of this task constraint in volleyball [33]. Thus, it is possible that the participants in our study managed to alter their jumping mechanics in order to generate similar performance outcomes. In fact, Gathercole et al. [34] reported that both kinetic (i.e., power production) and kinematic (i.e., contact time) parameters exhibited greater performance decrement, with only trivial changes in jump height during a countermovement jump protocol for up to 72 h following a fatiguing exercise. This phenomenon has been confirmed by a recent study [35], where kinetic parameters during a CMJ protocol were highly dependent on the training background of athletes (i.e., endurance and anaerobic athletes), and independent of jump performance, per se. Therefore, it is likely that our athletes altered their jumping strategy to maintain jump height, although further research examining kinetic parameters of jumping tasks during periods of EIMD is warranted in anaerobically-trained athletes.

As opposed to the CMJ and VJRUN, agility performance was impaired following the bout of RT. This discrepancy between VJRUN and agility performance may be due to differences in the complexity of the task constraints. For example, our participants were familiar with agility tasks, as change of direction capabilities are essential in volleyball [23], similar to that of jumping tasks. However, they may not have been able to strategically alter their technique with the agility protocol during the period of EIMD, as change of direction is a complex task requiring synchronization of several body parts [36], compared to that of linear sprinting and vertical jump. In fact, other studies have also shown impaired agility performance during periods of EIMD amongst elite athletes that require high levels of agility, including female basketball players [37] and male soccer players [38]. The potential mechanism for this decrement in change of direction speed has been postulated due to a compromise in lower body muscular function [37,38]. Indeed, lower body muscular strength and power have been correlated with measures of several agility protocols [39,40], demonstrating that exercise-induced reduction in muscular function may cause detrimental effects on agility performance.

Interestingly, de Freitas et al. [20] recently reported that EIMD caused no differences in agility performance in elite, male volleyball players. There may be several reasons for the discrepancy between the current findings and those by de Freitas et al. [20]. Firstly, de Freitas et al. [20] conducted their study in the second week of the pre-season, and reported that their participants had already commenced training. Therefore, this previous exposure to RT may have minimized potential muscle-damaging effects on agility performance due to the repeated bout effect [32]. In fact, DOMS in the lower body was twofold greater in our study (~5 out of 10), when compared to participants in the study by de Freitas et al. [20] (~2.25 out of 10) during the same period of EIMD (i.e., 24–48 h post-exercise). Secondly, de Freitas et al. [20] only included six participants, which the authors acknowledged was the limiting factor of their study. Conversely, the current study had almost threefold the number of participants (*n* = 16), which may have provided sufficient statistical power to detect differences between time points. Despite equivocal findings with literature in elite volleyball players, our findings suggest that the initial RT bout compromises agility performance for up to 48 h post-exercise following several weeks of off-season-related deconditioning.

With respect to the spiking-specific performance measures, the RT bout reduced ball speed whilst concomitantly improving accuracy. This decrement in speed at the cost of a heightened accuracy can be explained according to Fitt’s theory of the speed–accuracy trade-off [41] during throwing and kicking tasks in sport. However, the investigation of such a phenomenon during periods of EIMD has not been thoroughly investigated previously as far as we are aware, especially in volleyball. Rota et al. [42] reported that a strenuous tennis-specific exercise reduced ball speed and accuracy during a tennis serve, although the authors suspected that their participants may have reduced serve velocity with the intent of ameliorating decrement in serve accuracy. It has also been shown that individuals proactively alter preparatory state of movement and reduce movement speed during a mentally fatigued state, in order to preserve task success. Guerin and Kunkle [43] described how task constraints, such as spiking performance, could emerge and decay over time, given that constraints may mediate as “rate limiters” to improve performance. Therefore, it is possible that our participants may have adapted their motor behavior in order to overcome “rate limiters” (e.g., reduced ball speed due to the feeling of DOMS) by focusing on accuracy for task success.

## 5. Conclusions

The current study demonstrated that a bout of RT induced DOMS, impaired agility performance, and reduced ball speed during a spike-specific protocol for 48 h post-exercise in elite, female athletes returning from the off-season. Subsequently, athletes deconditioned from several weeks of RT (e.g., from the off-season or injury) may require more than 48 h of recovery following their initial bout of moderate-to-high intensity (~75% of 1RM) RT. This is particularly important if the RT bout is implemented prior to an important volleyball match (e.g., returning from injury), or a conditioning session involving high-intensity multidirectional exercises. In addition, athletes in the off-season are recommended to undertake at least one resistance training session a week at similar intensity and volume to their competitive season, where possible, to mitigate high levels of EIMD from their initial RT session in the pre-season. Studies have also reported accelerated recovery following strenuous training sessions, including massage, warm-down [44,45], and cold-water immersion [46], which warrants further research to determine whether such interventions may minimize the impact of EIMD on volleyball-specific performance measures.

## Figures and Tables

**Figure 1 ijerph-17-06448-f001:**
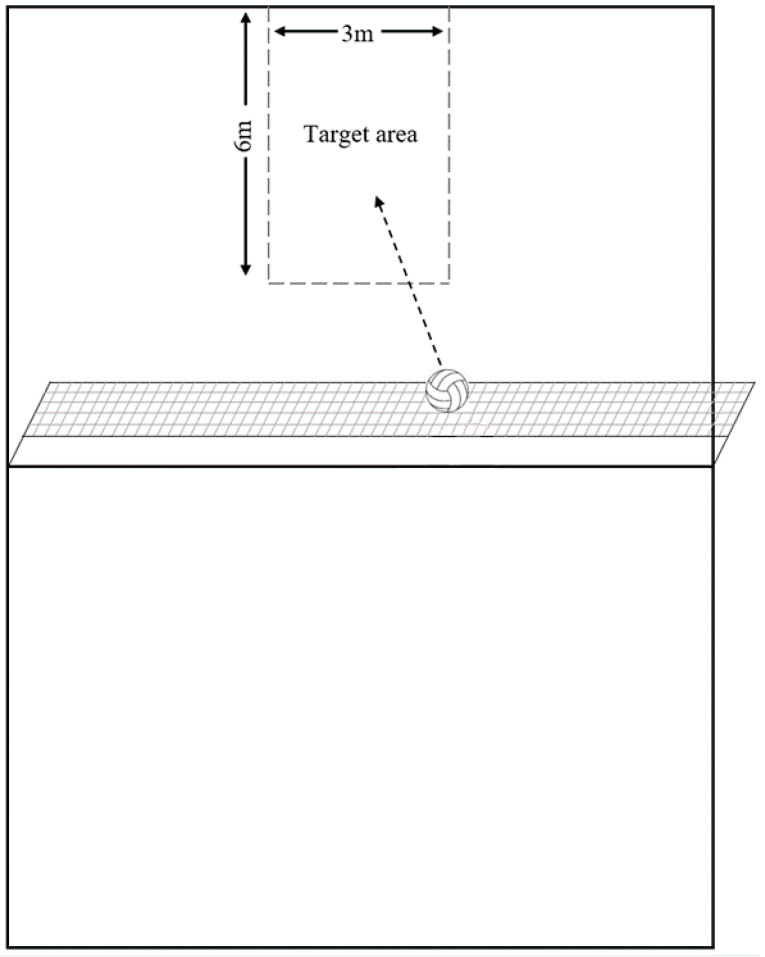
The schematic of the volleyball-specific performance protocol with the court shown from the top-view and the dotted grey lines displaying the target area.

**Figure 2 ijerph-17-06448-f002:**
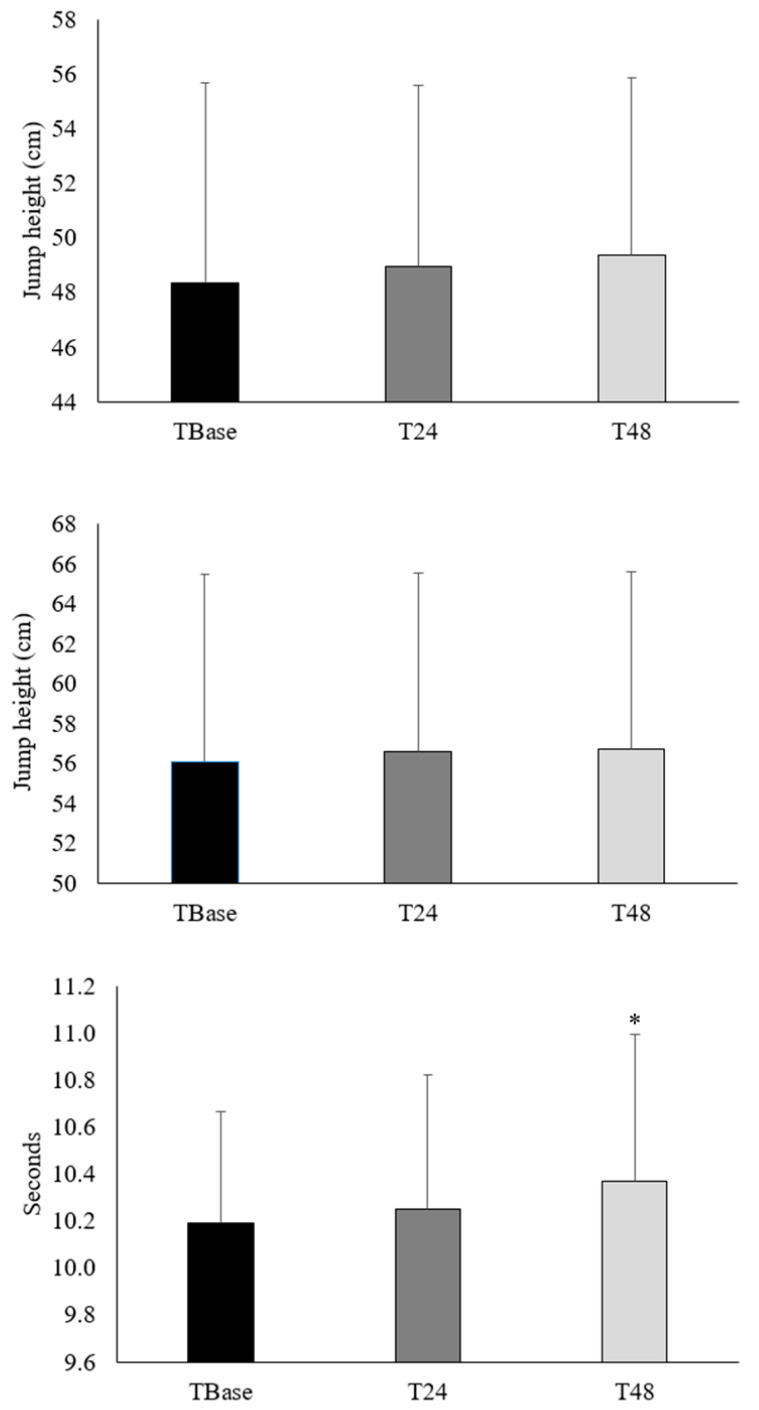
The sport-specific performance measures for countermovement jump (top), run-up vertical jump (middle), and T-agility performance protocol (bottom) between baseline (TBase) and 24 h post resistance training (RT) (T24) above the bar, between TBase and 48 h post RT (T48) above the bar, and between T24 and T48 in-between the bars. * Significantly different from TBase (*p* < 0.05).

**Figure 3 ijerph-17-06448-f003:**
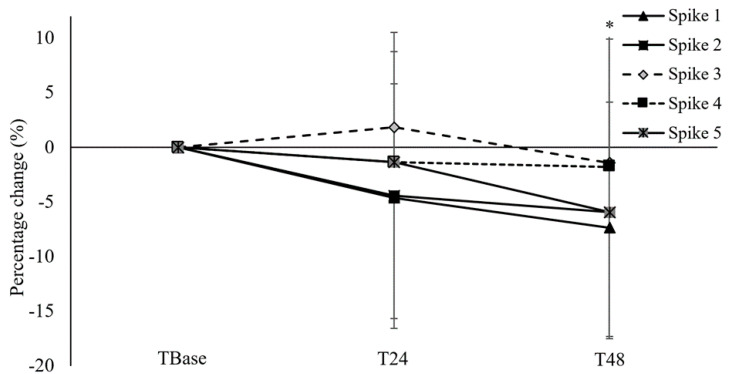
The percentage change in ball speed during the spiking performance test at 24 h (T24) and 48 h (T48) following the RT bout. * Significantly less than TBase as a main effect between days (*p* < 0.05).

**Table 1 ijerph-17-06448-t001:** The mean ± standard deviation of the indirect muscle damage markers for delayed onset of muscle soreness (DOMS) and sit-and-reach (SR) at baseline (TBase), 24 h (T24), and 48 h (T48) after the RT bout.

Measures	T_Base_	T_24_	T_48_
DOMS_Sqt_	1.94 ± 0.77	4.44 ± 1.79 *	5.25 ± 1.65 *
DOMS_Quad_	1.31 ± 0.70	4.75 ± 2.14 *	4.81 ± 2.76 *
DOMS_Gts_	1.19 ± 0.40	4.44 ± 2.52 *	4.38 ± 2.53 *
DOMS_Tri_	1.50 ± 0.73	4.13 ± 2.45 *	3.81 ± 2.10 *
DOMS_Abs_	1.19 ± 0.40	3.25 ± 2.14 *	3.69 ± 2.41 *
SR	49.1 ± 9.5	52.8 ± 8.5	53.0 ± 8.1

DOMSSqt—overall delayed onset of muscle soreness of the lower body; DOMSQuad—delayed onset of muscle soreness in the quadriceps; DOMSGts—delayed onset of muscle soreness in the gluteal muscles; DOMSTri—delayed onset of muscle soreness in the triceps; DOMSAbs—delayed onset of muscle soreness of the abdominal muscles. * Significantly different from TBase (*p* < 0.05).

**Table 2 ijerph-17-06448-t002:** The effect size calculations (95% confidence intervals) between baseline (T_Base_), 24 h (T_24_), and 48 h (T_48_) after the RT bout for delayed onset of muscle soreness measures and sit-and-reach (SR).

Measures	T_Base_-T_24_	T_Base_-T_48_	T_24_-T_48_
DOMS_Sqt_	1.81 (0.95–2.59) ^†^	2.57 (1.58–3.43) ^†^	0.47 (−1.16–0.24)
DOMS_Quad_	2.16 (1.24–2.97) ^†^	1.74 (0.89–2.50) ^†^	0.02 (−0.72–0.67)
DOMS_Gts_	2.25 (1.32–3.07) ^†^	1.76 (0.91–2.53) ^†^	0.03 (−0.67–0.72)
DOMS_Tri_	1.45 (0.64–2.19) ^†^	1.47 (0.66–2.21) ^†^	0.14 (−0.56–0.83)
DOMS_Abs_	1.34 (0.54–2.07) ^†^	1.45 (−0.64–2.19) ^†^	0.19 (−0.88–0.51)
SR	0.41 (−1.10–0.30)	0.44 (−1.13–0.27)	0.02 (−0.72–0.67)

DOMS_Sqt_ —overall delayed onset of muscle soreness of the lower body; DOMS_Quad_—delayed onset of muscle soreness in the quadriceps; DOMS_Gts_—delayed onset of muscle soreness in the gluteal muscles; DOMS_Tri_—delayed onset of muscle soreness in the triceps; DOMS_Abs_—delayed onset of muscle soreness of the abdominal muscles. ^†^ Large effects.

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
