# Peer review of "Resistance Training Acutely Impairs Agility and Spike-Specific Performance Measures in Collegiate Female Volleyball Players Returning from the Off-Season"

_ijerph, 2020, doi:10.3390/ijerph17186448_

Round 1

Reviewer 1 Report

Congratulations to the authors on their study. Please find some comments I believe will help improve the manuscript.

Abstract

-        Remove word ‘qualities’ from first sentence.

-        Add S.D. of % change in performance in results

-        Add effect sizes to beside p values

Intro

-        “.. involve sprinting…” are these not maximal accelerations as the constraints of court size would not enable an actual sprint.

-        Surely RT is undertaken to develop force-velocity characteristics of performance as opposed to meeting “the anaerobic demands of the sport”? This makes it sound like RT is performed from a metabolic point-of-view

-        Early in the intro I would introduce EIMD as part of range of potential mechanisms that are potentially detrimental to performance following recovery from RT, and provide a few other examples.

-        “increased intramuscular proteins..” its not that they increase in number per se, but they appear in the likes of blood serum in higher numbers than usual i.e. they increase in a particular location greater than previously.

-        Much more details needed in description of your custom-made device for CMJ measurement.

Methods

-        2.2. The participants are described as ‘elite’, does this mean national representation or how exactly is the population defined as elite? Please add for clarity.

-        2.2. Is it possible to have some sort of quantification of the ‘reduced volume’ of RT completed this is quite vague. If not a quantification but even a description?

-        2.3. Please remove CMJ as an ‘indirect marker of muscle damage’. It could be put with sport-specific tasks surely? I.e. a CMJ is performed during a standing block in volleyball?

-        2.3. Not clear here on the DOMS – are these perceptions of DOMS? Did the athletes rate perception of DOMS following a contraction of the muscle groups? This needs to be made clear. If so, what was the scale or how did they rate DOMS?

-        2.4 For VJRUN what was the distance between pegs i.e. sensitivity.

-        2.4 Not sure Fig 1 is needed as the T-test is virtually known everywhere I presume?

-        2.5 was the set always of sufficient quality to allow a maximal spike effort? I.e. were any trials disregarded as the set wasn’t optimal? May be of use just to state this either way.

-        2.7 please include the inter-set rest periods for exercises.   

-        2.8 can the authors please report the effect size for the ANOVA’s main interaction effects in addition to the pairwise Cohen D’s (such as partial eta squared)

-        2.8 Im not sure of the value of comparing between muscle groups over and above the usefulness of the within group change. Knowing for example the triceps have more DOMS than quads, I don’t see the practical application of this or the relationship. If both had significant decrements from baseline then this would be more important to inform a recovery focus or intervention than knowing a difference between the groups.

Results

-        When reporting % differences please include the S.D.

Discussion

-        “…no anaerobic training background…” change to RT background. Runners will have an anaerobic background to some degree. Same for endurance and anaerobic athletes, change to resistance trained

Author Response

Reviewer’s comment: Congratulations to the authors on their study. Please find some comments I believe will help improve the manuscript.

Author’s response: Dear reviewer, thank you very much for your comments. We believe that your thoughtful feedback has improved the quality of our work.

Abstract

Reviewer’s comment: Remove word ‘qualities’ from first sentence.

Author’s response: Removed

Reviewer’s comment: Add S.D. of % change in performance in results

Author’s response: Included

Reviewer’s comment: Add effect sizes to beside p values

Author’s response: Included

Introduction

Reviewer’s comment: “.. involve sprinting…” are these not maximal accelerations as the constraints of court size would not enable an actual sprint.

Author’s response: Thank you, we have reworded this to ‘maximal accelerations’

Reviewer’s comment: Surely RT is undertaken to develop force-velocity characteristics of performance as opposed to meeting “the anaerobic demands of the sport”? This makes it sound like RT is performed from a metabolic point-of-view

Author’s response: We have reworded the sentence to ‘Thus, volleyball players are commonly required to undertake resistance training (RT) to develop force-velocity characteristics of performance’.

Reviewer’s comment: Early in the intro I would introduce EIMD as part of range of potential mechanisms that are potentially detrimental to performance following recovery from RT, and provide a few other examples.

Author’s response: Potential mechanism of EIMD, and how this results to force loss, has now been included earlier in the Introduction.

Reviewer’s comment: “increased intramuscular proteins..” its not that they increase in number per se, but they appear in the likes of blood serum in higher numbers than usual i.e. they increase in a particular location greater than previously.

Author’s response: We agree that our wording was incorrect. We have reworded this to ‘increased leakage of intramuscular proteins into the systemic circulatory system’.

Reviewer’s comment: Much more details needed in description of your custom-made device for CMJ measurement.

Author’s response: Thank you for the suggestion. We have provided further description of the custom-built vertical jump apparatus in the methods section.

Methods

Reviewer’s comment: 2.2. The participants are described as ‘elite’, does this mean national representation or how exactly is the population defined as elite? Please add for clarity.

Author’s response: We have excluded the term ‘elite’ to avoid confusion. To align this amendment with the title, we have also reworded ‘elite’ to ‘collegiate’.

Reviewer’s comment: 2.2. Is it possible to have some sort of quantification of the ‘reduced volume’ of RT completed this is quite vague. If not a quantification but even a description?

Author’s response: We have included that the participants reduced their RT volume by at least half.

Reviewer’s comment: 2.3. Please remove CMJ as an ‘indirect marker of muscle damage’. It could be put with sport-specific tasks surely? I.e. a CMJ is performed during a standing block in volleyball?

Author’s response: We agree, and thus CMJ has now been included as part of sport-specific tasks. This change has also been accommodated for the Results section.

Reviewer’s comment: 2.3. Not clear here on the DOMS – are these perceptions of DOMS? Did the athletes rate perception of DOMS following a contraction of the muscle groups? This needs to be made clear. If so, what was the scale or how did they rate DOMS?

Author’s response: The description of how DOMS was recorded has now been included.

Reviewer’s comment: 2.4 For VJRUN what was the distance between pegs i.e. sensitivity.

Author’s response: As per the description expanded in the methods, the distance between pegs was one centimeter.

Reviewer’s comment: 2.4 Not sure Fig 1 is needed as the T-test is virtually known everywhere I presume?

Author’s response: We agree, Fig 1 has now been excluded.

Reviewer’s comment: 2.5 was the set always of sufficient quality to allow a maximal spike effort? I.e. were any trials disregarded as the set wasn’t optimal? May be of use just to state this either way.

Author’s response: To ensure ecological validity, the participants were not allowed extra attempts, irrespective of the quality of each set. This has now been included in the Methods section.

Reviewer’s comment: 2.7 please include the inter-set rest periods for exercises.  

Author’s response: Approximately two minutes of rest was provided in-between each set.

Reviewer’s comment: 2.8 can the authors please report the effect size for the ANOVA’s main interaction effects in addition to the pairwise Cohen D’s (such as partial eta squared)

Author’s response: Effect size calculations has now been included for both interaction and main effects of the repeated measures ANOVA.

Reviewer’s comment: 2.8 Im not sure of the value of comparing between muscle groups over and above the usefulness of the within group change. Knowing for example the triceps have more DOMS than quads, I don’t see the practical application of this or the relationship. If both had significant decrements from baseline then this would be more important to inform a recovery focus or intervention than knowing a difference between the groups.

Author’s response: We agree, and thus, interaction effects of day x muscle has been excluded, and DOMS has now been analysed using a one-way repeated measures ANOVA.

Results

Reviewer’s comment: When reporting % differences please include the S.D.

Author’s response: Thank you, S.D. has been included both in text and in the figure. For the accuracy, the percentage is reported as a frequency (i.e., proportion), rather than percentage change, and thus, no S.D. is required for this dichotomous variable.

Discussion

Reviewer’s comment: “…no anaerobic training background…” change to RT background. Runners will have an anaerobic background to some degree. Same for endurance and anaerobic athletes, change to resistance trained

Author’s response: We agree, and we have now changes the text to RT background.

Reviewer 2 Report

The authors present a thoroughly designed study, examining a homogenous cohort of sixteen female volleyball players. The introduction offers an adequate insight into the current state of knowledge. More information on the recovery-phase after completion of the RT program would be useful. Several studies have discussed e.g massages and cold-water immersions as promising interventions for the reduction of DOMS (Bale P, James H.Massage, warmdown and rest as recuperative measures after short term intense exercise. Physiotherapy in Sport 1991;13:4–7.; Smith LL, Keating MN, Holbert D, et al. The effects of athletic massage on delayed onset muscle soreness, creatinekinase, and neutrophil count: a preliminary report. JOSPT1994;19:93–9.; Tiidus PM, Shoemaker JK. Effleurage massage, muscleblood flow and long-term post-exercise strength recovery.Int J Sports Med 1995;16:478–83.; Bleakley C, McDonough S, Gardner E, Baxter GD, Hopkins JT, Davison GW. Cold-water immersion (cryotherapy) for preventing and treating muscle soreness after exercise. Cochrane Database Syst Rev. 2012;2012(2):CD008262. Published 2012 Feb 15. doi:10.1002/14651858.CD008262.pub2)

The references show a large number of self-citations, primarily by the first author:

4. Doma, K. and G. Deakin, The Acute Effect of Concurrent Training on Running Performance Over 6 Days. Res Q Exerc Sport 2015, 86, 387-96.

6. Doma, K., G.B. Deakin, and D.J. Bentley, Implications of Impaired Endurance Performance following Single Bouts of Resistance Training: An Alternate Concurrent Training Perspective. Sports Med 2017, 47, 2187-2200.

7. Doma, K., et al., The repeated bout effect of traditional resistance exercises on running performance across 3 bouts. Appl Physiol Nutr Metab 2017, 42, 978-985.

8. Doma, K., et al., The repeated bout effect of typical lower body strength training sessions on sub-maximal running performance and hormonal response. Eur J Appl Physiol 2015, 115, 1789-99.

12. Doma, K., et al., Training Considerations for Optimising Endurance Development: An Alternate Concurrent Training Perspective. Sports Med 2019, 49, 669-682.

16. Doma, K. and G.B. Deakin, The effects of strength training and endurance training order on running economy and performance. Appl Physiol Nutr Metab 2013, 38, 651-6.

17. Doma, K. and G.B. Deakin, The acute effects intensity and volume of strength training on running performance. Eur J Sport Sci 2014, 14, 107-15.

18. Doma, K., et al., The Effect of a Resistance Training Session on Physiological and Thermoregulatory Measures of Sub-maximal Running Performance in the Heat in Heat-Acclimatized Men. Sports Med Open 2019, 5, 21.

24. Nagata, A. and T. Fuchimoto, The development of a method to measure the maximum spike height in volleyball. J Volleyball Sci 2011, 13, 1-7.

34. Doma, K., et al., Impact of Exercise-Induced Muscle Damage on Performance Test Outcomes in Elite Female Basketball Players. J Strength Cond Res 2018, 32, 1731-1738.

On the other hand further literature could be discussed, e.g. Naclerioet al (2013), who also partially examined female volleyball college players (Naclerio F, Faigenbaum AD, Larumbe-Zabala E, et al. Effects of different resistance training volumes on strength and power in team sport athletes. J Strength Cond Res. 2013;27(7):1832-1840. doi:10.1519/JSC.0b013e3182736d10).

The manuscript is well written. English language and style are fine.

Author Response

Reviewer’s comment: The authors present a thoroughly designed study, examining a homogenous cohort of sixteen female volleyball players. The introduction offers an adequate insight into the current state of knowledge. More information on the recovery-phase after completion of the RT program would be useful. Several studies have discussed e.g massages and cold-water immersions as promising interventions for the reduction of DOMS (Bale P, James H.Massage, warmdown and rest as recuperative measures after short term intense exercise. Physiotherapy in Sport 1991;13:4–7.; Smith LL, Keating MN, Holbert D, et al. The effects of athletic massage on delayed onset muscle soreness, creatinekinase, and neutrophil count: a preliminary report. JOSPT1994;19:93–9.; Tiidus PM, Shoemaker JK. Effleurage massage, muscleblood flow and long-term post-exercise strength recovery.Int J Sports Med 1995;16:478–83.; Bleakley C, McDonough S, Gardner E, Baxter GD, Hopkins JT, Davison GW. Cold-water immersion (cryotherapy) for preventing and treating muscle soreness after exercise. Cochrane Database Syst Rev. 2012;2012(2):CD008262. Published 2012 Feb 15. doi:10.1002/14651858.CD008262.pub2)

Author’s response: Thank you for your comments, and for assisting us in improving the quality of our work. We have now cited some of the above-mentioned studies in the Discussion as potential future studies. We believe including these studies in the Introduction may cause confusion for readers, as the primary aim of the studies were to examine the acute effects of RT on sport-specific performance.

Reviewer’s comment: The references show a large number of self-citations, primarily by the first author

Author’s response: Thank you for pointing this out. We have now excluded three studies and replaced these with work by others.

Reviewer’s comment: On the other hand further literature could be discussed, e.g. Naclerioet al (2013), who also partially examined female volleyball college players (Naclerio F, Faigenbaum AD, Larumbe-Zabala E, et al. Effects of different resistance training volumes on strength and power in team sport athletes. J Strength Cond Res. 2013;27(7):1832-1840. doi:10.1519/JSC.0b013e3182736d10).

Author’s response: This is a very interesting study, and thank you for suggesting this study. However, given that the study incorporated athletes from different sporting backgrounds and gender, the intervention was for chronic adaptations rather than acute responses and that the outcome measures were not sport-specific, we believe this study may not be applicable for this manuscript.

Reviewer’s comment: The manuscript is well written. English language and style are fine.

Author’s response: Thank you.